# Influence of Sc Microalloying on the Microstructure of Al5083 Alloy and Its Strengthening Effect

Ji-Hoon Park [1,2], Kee-Ahn Lee [2], Sung-Jae Won [1], Yong-Bum Kwon [1] and Kyou-Hyun Kim [1,*]

1   Korea-Russia Innovation Center, Korea Institute of Industrial Technology, Incheon 21999, Korea; chco2002@kitech.re.kr (J.-H.P.); bluetjdwo@kitech.re.kr (S.-J.W.); kyb916@kitech.re.kr (Y.-B.K.)
2   Department of Advanced Materials Engineering, In-Ha University, Incheon 22212, Korea; keeahn@inha.ac.kr
*   Correspondence: khkim1308@kitech.re.kr; Tel.: +82-10-6240-5759

**Abstract:** In this study, we investigate the influence of Sc microalloying on the microstructure of the Al5083 alloy. Trace amounts of Sc addition drastically improve the mechanical properties of the Al5083 alloy from 216 MPa to 233 MPa. Macroscopically, the addition of Sc significantly reduces the grain size of Al by approximately 50%. Additionally, a microstructural investigation reveals that the Sc microalloying element induces fine $Al_3Sc$ nanoprecipitates in the Al matrix. The formation of $Al_3Sc$ nanoprecipitates results in a pinning effect on the dislocations, leading to accumulated dislocations. Compared to a Sc-free Al5083 alloy specimen, the number density of dislocations in the Sc-added Al5083 alloy significantly increases after hot rolling, enhancing the tensile properties. We reveal that the improved mechanical properties of Al5083 with Sc microalloying originate from the grain refinement and the formation of fine $Al_3Sc$ nanoprecipitates.

**Keywords:** Al 5xxx alloy; Al5083 alloy; microstructure; microalloying





## 1. Introduction

The Al-Mg-based Al5083 alloy has attracted much research interest due to its broad range of applications to structural components in the transportation and construction industries given its high strength, ductility, corrosion resistance, and weldability [1]. For the Al5083 alloy, an intermediate amount of Mn (~5 wt.%) is utilized in order to enhance the mechanical properties. Fe, the most typical impurity in Al alloys, is known to appear as plate-like, Fe-rich intermetallic particles in the grain boundaries [2], deteriorating the mechanical properties. Elemental Mn then changes the plate-like, Fe-rich intermetallic particles into skeleton-like particles, which, in turn, improves the mechanical properties of the Al5083 alloy [2,3].

Thermomechanical processes such as hot rolling have been widely used to improve the mechanical properties of Al alloys. During hot rolling, strain is applied to Al alloys, increasing the dislocation density and consequently improving the mechanical properties due to the work hardening effect [4,5]. In addition, hot rolling breaks the large and brittle particles formed in Al alloys into fine particles. In consequence, the ductility of cast Al alloys can be improved by hot rolling.

In addition to the thermomechanical process, Sc microalloying has often been introduced to improve the mechanical properties of Al alloys [6]. The microstructure and mechanical properties are significantly affected even by a small addition of the Sc microalloying element [7,8]. In Al-Mg-based Al alloys, it has been reported that the existence of Mg strongly reacts with Sc atoms [9]. The strong reaction between the Mg and Sc atoms then affects the aging response of Sc-microalloyed Al-Mg-based alloys. The addition of Sc forms $Al_3Sc$ precipitates in the Al matrix. The diffusivity of Mg atoms is then severely restricted by the $Al_3Sc$ precipitates [9]. This consequently reduces the growth rate of the $Al_3Sc$ precipitate, which thus enhances the creep resistance of Al-Mg-based alloys. Additionally, finely dispersed $Al_3Sc$ precipitates reportedly inhibit recrystallization during severe plastic

deformation and during the annealing process [10,11]. Strain hardening can then be used to improve the mechanical strength of Sc-containing Al-Mg-based alloys without the annealing process commonly used for conventional heat-treated Al alloys [7]. In addition, the existence of $Al_3Sc$ particles in the melt act as nuclei for $\alpha$-Al during the solidification step, leading to significant grain refinement [12]. Nevertheless, the effect of Sc microalloying has been in general reported for macroscopic scale. It is therefore the microstructural evolution at nanoscopic scale should be carefully considered to understand the improvement of mechanical properties by the trace addition of Sc microalloying element.

In this study, we aimed to investigate the effect of trace addition of Sc on the microstructural evolution of the Al5083 alloy. The microstructural evolution due to the addition of Sc is also expected to affect the hot-rolling process as a thermomechanical treatment for work hardening. The microstructures of the as-cast Sc-free/microalloyed Al5083 alloys were investigated from the macroscopic scale to the microscopic scale using X-ray diffraction (XRD), optical microscopy (OM), secondary electron microscopy (SEM), and transmission electron microscopy (TEM). The resultant microstructural observations analyzed considering the mechanical properties of the fabricated Al alloys.

## 2. Experimental Procedures

Al5083 was used as a base composition in this study, and the corresponding chemical compositions are presented in Table 1. Al-5 wt.% Mn, Al-5 wt.% Ti, Al-5 wt.% Fe, and Al-2 wt.% Sc master alloys were used, with approximately 99.9 wt.% pure elements used for the other components. The elements were initially melted using a high-frequency induction furnace at 800 °C under atmospheric conditions. Ar gas was then injected into the molten metal for 15 min to enhance the homogeneity and to remove any impurities from the molten metal. After the gas bubbling treatment, the molten metal was poured into a mold at 750 °C. The as-cast samples were then homogenized at 475 °C for 24 h in order to improve the workability of the alloy during the hot working process. The homogenized samples were also hot-rolled up to 60% at 450 °C and stabilized at 150 °C for 2 h to reduce the internal energy induced by hot rolling.

**Table 1.** Chemical compositions of the fabricated Sc-free/microalloyed Al5083 alloys.

| Element (wt.%) | Si | Fe | Cu | Mn | Mg | Cr | Zn | Ti | Sc | Al |
|---|---|---|---|---|---|---|---|---|---|---|
| Sample A | 0.4 | 0.4 | 0.1 | 0.5 | 4.9 | 0.25 | 0.25 | 0.15 | - | Bal. |
| Sample B | 0.4 | 0.4 | 0.1 | 0.5 | 4.9 | 0.25 | 0.25 | 0.15 | 0.15 | Bal. |

For the mechanical test, the specimens were prepared according to ASTM E8 standard test methods with the gage length, total length, width, and thickness of 25 mm, 100 mm, 6.0 mm, and 2.5 mm, respectively. Tensile tests were then carried out using a universal testing machine (UTM, Shimadzu, AG-300kNX Plus, Kyoto, Japan) with a constant strain rate of 2.0 mm/min at room temperature. More than 10 specimens were measured to obtain average values of mechanical properties.

The macroscopic structure was investigated via optical microscopy (OM, Leica DM750M, Wetzlar, Germany) with ImageJ software to quantify the grain size. In addition, the overall microstructure was characterized based on X-ray diffraction (XRD, Bruker Corp., D8 Advance, Hanau, Germany). The microscopic structure was then examined using scanning electron microscopy (SEM, Jeol Ltd., JSM-7100F, Tokyo, Japan). For the SEM investigation, the specimens were mechanically polished using standard metallographic techniques and etched by Keller's reagent [13]. Additionally, the nanoscopic structure was studied via field-emission transmission electron microscopy (TEM, Tecnai ST-F20, FEI, Eindhoven, The Netherlands). For electron transparency, each sample was initially polished down to <20 μm and Ar-ion milled at an incident ion beam angle of 6° with an accelerating voltage of 4 kV.

## 3. Results and Discussion

Figure 1 shows the typical tensile stress–strain curves obtained from the developed Al-Mg-(Sc)-based alloys with different post-treatment processes. The tensile properties of the yield strength ($\sigma_y$), ultimate tensile strength ($\sigma_{UTS}$), and total elongation ($\varepsilon_f$) are also given in Table 2. The tested specimens are categorized into two major groups with/without the trace addition of Sc (samples A and B) and with/without the additional hot-rolling processes (samples A′ and B′). Thus, the effect of the addition of Sc can be investigated by comparing samples A and B. From the tensile test results, the tensile strength ($\sigma_{UTS}$) is improved from 216 MPa to 233 MPa by the addition of a small amount of Sc (0.15 wt.%). Meanwhile, the elongation remains nearly identical at approximately 4% regardless of the addition of Sc. In order to determine the effect of hot rolling, the tensile properties were then compared based on samples A′ and B′. As presented in Table 2, the additional process resulted in the simultaneous improvement in both the tensile strength and the elongation. The $\sigma_{UTS}$ value is increased to 304 MPa for sample A′ and 330 MPa for sample B′ due to the hot-rolling process. Moreover, the elongation is significantly improved from ~4% to ~12% for both samples. In addition to the elongation outcomes of samples A and B, the elongation results of samples A′ and B′ are nearly identical even after the hot-rolling process. It is generally known that hot rolling causes an increase in the number of internal defects such as dislocations and twin grains in specimens, along with deformation of the grains, which affects the mechanical strength [4,5]. From the tensile results, the highest tensile strength of Al-Mg-Mn-based alloys can be realized with a trace addition of Sc and an additional hot-rolling process. The effects of adding Sc and the hot-rolling process were initially investigated based on a macroscopic structural investigation using XRD.

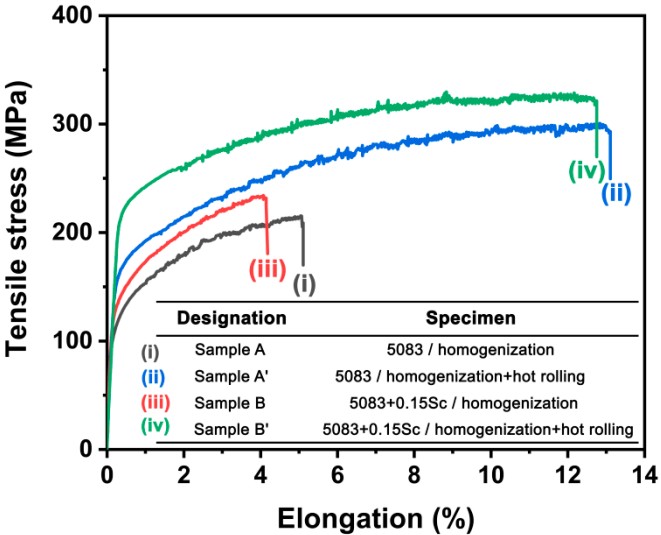

**Figure 1.** Tensile properties of the Al5083 alloy (sample A) and Sc-microalloyed Al5083 (sample B) alloy from the tensile tests of hot-rolled sample A (sample A′) and sample B (sample B′).

Figure 2 shows the two-theta scan profiles when using XRD obtained from samples A/A′ and B/B′. The XRD profiles consist of strong diffraction peaks with several weak diffraction peaks. First, the strong diffraction peaks for samples A/A′ and B/B′ are all identified as Al, as shown in Table 3. The experimental Al peaks, however, are slightly shifted to the left in 2θ. This indicates that the samples are tensile-strained. Second, the weak peaks are determined as $Mg_2Si$, $Al_8Mn_5$, and $Al_6$(Fe, Mn) phases based on the reported crystallographic information [14–17]. In contrast, a Sc-related phase such as the $Al_3M′$ type phase is not found in the Sc-added Al5083 alloy (sample B/B′). It is considered that the $Al_3M′$ type phase is very fine size [18–20] such that XRD is incapable of detecting Sc-related precipitates due to the relatively large beam probe size [21]. Details will be discussed with the TEM investigation.

**Table 2.** Tensile properties of Sc-free/microalloyed Al5083 alloys with/without the hot-rolling process.

| Designation | Specimen | $\sigma_y$ (MPa) | $\sigma_{UTS}$ (MPa) | $\varepsilon$ (%) |
|---|---|---|---|---|
| Sample A | 5083/ homogenization | 113 | 216 ± 9 | 4.7 ± 1.2 |
| Sample A′ | 5083/ homogenization + hot rolled | 183 | 304 ± 13 | 12.5 ± 1.1 |
| Sample B | 5083 + 0.15Sc/ homogenization | 135 | 233 ± 9 | 4.1 ± 0.8 |
| Sample B′ | 5083 + 0.15Sc/ homogenization + hot rolled | 214 | 330 ± 10 | 12.4 ± 1.1 |

Figure 3a,b shows typical optical micrographs (OM) recorded from the synthesized Al-Mg-Mn-(Sc)-based alloys. The homogenized Al-Mg-Mn-(Sc) alloys consist of uniformly distributed equiaxed grains, as shown in the magnified OM images (inset of Figure 3a,b). From the OM images, a trace amount of Sc addition induces a difference in the grain size, as shown in Figure 3a,b. The grain sizes of the synthesized alloys are measured and averaged using a postimage processing technique provided in the ImageJ software (1.52, National Institute of Mental Health, ML, USA) used here. The average grain size of the homogenized Al5083 alloy is ~83.99 μm. This average grain size is then drastically reduced to ~41.29 μm upon the addition of 0.15 wt.% Sc, which is only ~50% of the original grain size (Al5083). In Al-based alloys, it has been reported that Sc microalloying elements enhance the solidification speed due to the faster solidification phase transition caused by the supercooling effect [22]. Additionally, the Sc constituent is known to promote the formation of Al$_3$Sc precipitates, which hinder the grain growth of $\alpha$-Al [12]. Based on earlier work, the addition of Sc in Al alloys mainly causes grain refinement, thus increasing the mechanical strength. The microstructure of the hot-rolled specimens (samples A′ and B′) were then investigated, as shown in Figure 3c,d. Neither of the hot-rolled specimens shows obvious dynamic grain growth at the relatively high rolling temperature of 450 °C.

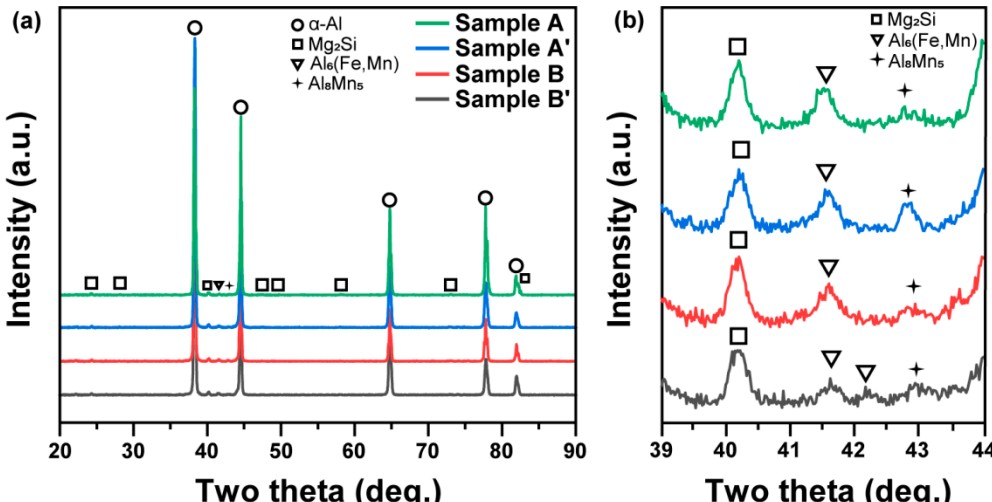

**Figure 2.** Phase identification of (**a**) the Sc-free/microalloyed Al5083 alloys using 2θ scans of XRD with (**b**) the magnified XRD profiles from 39°–44° for the phase identification of the precipitates.

**Table 3.** Experimental and calculated 2θ Angles with the corresponding differences for the Al phase.

| Al (Ref.) | | Sample A | | | Sample A′ | | | Sample B | | | Sample B′ | | |
|---|---|---|---|---|---|---|---|---|---|---|---|---|---|
| (h, k, l) | 2θ | (h, k, l) | 2θ | Δ2θ | (h, k, l) | 2θ | Δ2θ | (h, k, l) | 2θ | Δ2θ | (h, k, l) | 2θ | Δ2θ |
| 111 | 38.47 | 111 | 38.26 | −0.21 | 111 | 38.32 | −0.15 | 111 | 38.32 | −0.15 | 111 | 38.23 | −0.24 |
| 200 | 44.72 | 200 | 44.56 | −0.16 | 200 | 44.46 | −0.26 | 200 | 44.51 | −0.21 | 200 | 44.70 | −0.02 |
| 220 | 65.09 | 220 | 64.78 | −0.31 | 220 | 64.59 | −0.50 | 220 | 64.55 | −0.54 | 220 | 64.70 | −0.39 |
| 311 | 78.23 | 311 | 77.77 | −0.46 | 311 | 77.57 | −0.66 | 311 | 77.53 | −0.70 | 311 | 77.80 | −0.43 |
| 222 | 82.43 | 222 | 81.92 | −0.51 | 222 | 81.94 | −0.49 | 222 | 81.96 | −0.47 | 222 | 82.00 | −0.43 |

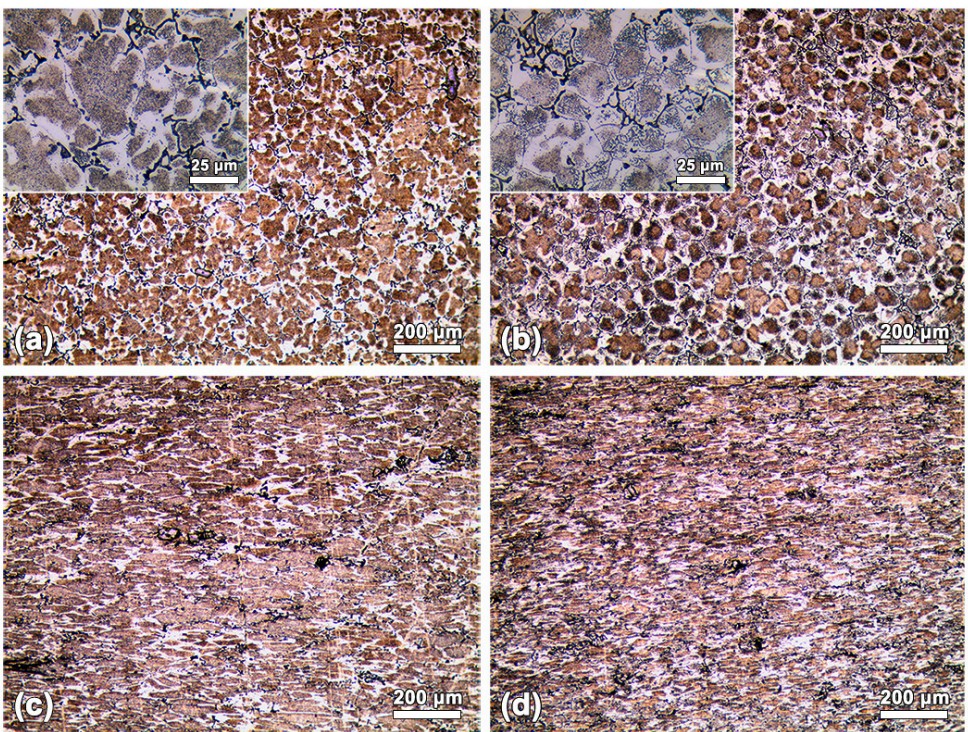

**Figure 3.** Typical OM images recorded from (**a**) sample A (Sc-free and non-hot-rolled), (**b**) sample B (Sc-microalloyed and non-hot-rolled), (**c**) sample A′ (Sc-free and hot-rolled), and (**d**) sample B′ (Sc-microalloyed and hot-rolled).

Figure 4 shows backscattered electron (BSE) images recorded from samples A and B with the corresponding EDS mapping results. From the recorded BSE images, the synthesized Sc-free and 0.15 Sc-added Al4.9Mg0.5Mn alloys have similar microstructures, as can be seen in Figure 4a,c. The Al4.9Mg0.5Mn-(0.15 Sc) alloys mainly consist of an Al matrix and three different phases in the grain boundaries, as indicated in Figure 4a. From the elemental mapping results, the large particle (I) is rich in the three elements of Fe, Cr, and Mn, which may have formed in the $Al_6M$ type phase, where M can be Fe/Cr/Mn [23,24]. The $Al_6M$-type phase is in good agreement with the phase identification result from XRD (Figure 2). Other particles with a skeleton-like structure (II) are simultaneously observed with $Al_6M$-type particles. The two elements of Cr and Mn show much brighter intensity levels, compared to the other elements. Based on previous reports, in consequence, the skeleton-like structure (II) is considered to be the Al-(Cr, Mn) type of intermetallic phase [24]. The particles with dark contrast (III) are then determined as $Mg_2Si$ after a chemical analysis and considering the XRD results (Figure 2). The Al-Mg phase diagram [25] shows that the solubility of Mg in Al is within 17% and that the phase transformation temperature to α-Al + $Al_8Mg_5$ from the Al-Mg alloy in the liquid state is approximately 723 K. It is therefore considered that the liquid state is mainly solidified to the α-Al phase without the formation of Al-Mg precipitates due to the insufficient Mg-rich

eutectic phase caused by the low content of Mg and the relatively insufficient temperature. On the other hand, the trace addition of Sc leads to no distinct formation of a Sc-containing secondary phase in the Al-4.9Mg-0.5Mn-0.15Sc alloy. This is due to the fact that Sc-related precipitates such as those with the $Al_3M'$ type structure are only a few nanometers in size, in the case of which M' could be Sc or a random solid solution with other elements in the Al alloys [26,27].

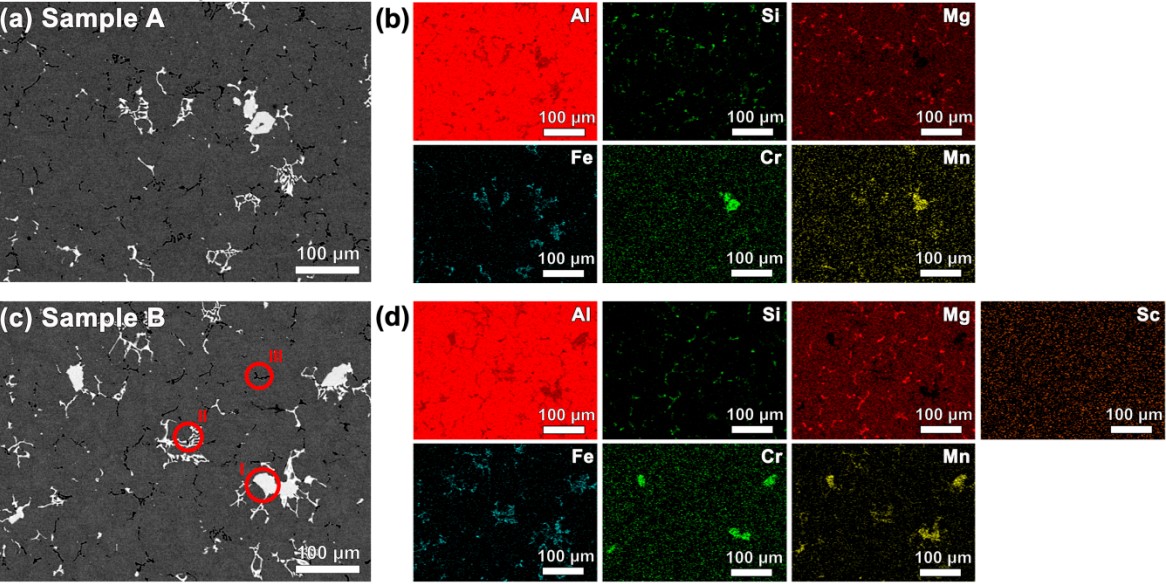

**Figure 4.** Typical BSE images of non-hot-rolled (**a**) sample A (Sc-free Al5083), (**c**) sample B (Sc-microalloyed Al5083) and (**b**,**d**) the corresponding elemental mapping results from SEM.

Figure 5 shows bright-field (BF) images and the elemental mapping results of samples A and sample B. The diffraction contrast images were recorded at the low-order zone axis of $[110]_{Al}$ in order to improve the diffraction contrast. Samples A and B show no obvious microstructural differences in the medium magnification images shown in Figure 5a,d. As shown in the inset of Figure 5a, the typical ED pattern of sample A only consists of Al spots without any secondary spots. In contrast, the ED pattern of sample B shows extra peaks which are indexed as the $Al_3M'$-type structure, as indexed in the inset of Figure 5d. The diffraction spots of the $Al_3M'$ phase exist at $\frac{1}{2}$ g of the Al reflections, which indicates that the lattice constants of Al and $Al_3M'$ are similar. Besides the formation of $Al_3M'$-type precipitates in sample B, the addition of a trace amount of Sc affects the number density of the precipitates in the $\alpha$-Al grains of samples A and B. Figure 5b,e shows Z-contrast images, respectively, recorded from sample A and B using a scanning TEM technique with a high angular annular dark-field (HAADF) detector. Both sample A and sample B consist of very fine nanoprecipitates in the $\alpha$-Al grains, while the number density of the nanoprecipitates significantly increases in the Sc-microalloyed Al5083 alloy. For samples A and B, the observed nanoprecipitates are mostly revealed as $Al_8Mn_5$ and $Al_6(Fe,Mn)$ from the elemental mapping results, as shown in Figure 5c,f. On the other hand, $Al_3Sc$ ($Al_3M'$ type) precipitates are additionally observed in sample B. From the microscopic observations, it is evident that the addition of Sc facilitates the formation of nanoprecipitates in the Al matrix. Interestingly, the effect of a Sc addition on the formation of precipitates is known to show different results in Al alloys, including (i) an enhancement [28,29], (ii) no effect [30,31], and (iii) suppression [32,33]. Nevertheless, the results of this study clearly show that the trace amounts of added Sc facilitate the formation of nanoprecipitates in the Al-Mg-Mn alloy. Based on the above results, it is considered that the increase in the number density of the nanoprecipitates contributes to the enhancement of the mechanical properties of the Sc-added Al5083 alloy.

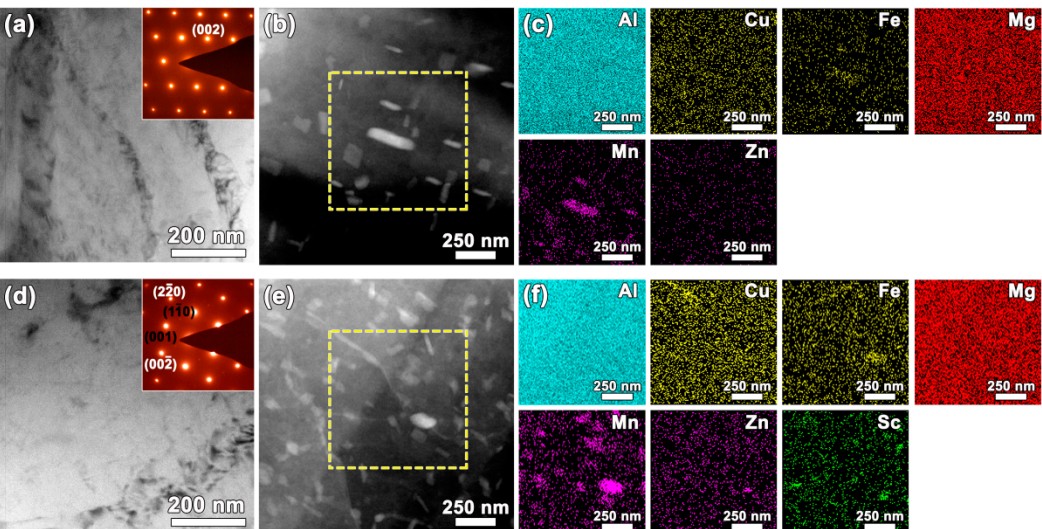

**Figure 5.** Microscopic structural investigation using TEM for (**a–c**) sample A and (**d–f**) sample B: (**a**,**d**) typical BF images with the corresponding electron diffraction patterns (insets) and (**b**,**e**) Z-contrast images with (**c**,**f**) the corresponding elemental mapping results.

Figure 6 shows bright-field (BF) images of the hot-rolled samples A′ and B′. The dislocation density is generally known to increase due to the hot-rolling process. In comparison to Figure 5, the dislocation density increases more in samples A′ and B′. In addition, as shown in Figure 6a,b, the addition of Sc clearly affects the dislocation density in the $\alpha$-Al grain. The Sc-free alloy of sample A′ (Figure 6a) has fewer dislocations, compared to the Sc-added alloy of sample B′ (Figure 6b). In the magnified BF images of samples A′ and B′, it can be observed that dislocations are well developed around the nanoprecipitates. Samples A′ and B′ are hot-rolled at an identical reduction rate of 60% such that the difference in the dislocation density likely originated from the microstructural difference induced by the addition of Sc. As shown in Figure 5, the addition of Sc facilitates the formation of nanoprecipitates in the Al matrix such as $Al_6M$ (M: Fe/Cr/Mn), $Al_3M'$ (M′: Sc/Ti), $Mg_2Si$, and other phases, which cannot be exactly identified. The size of nanoprecipitates was then measured by few nm ~ few tens nm. The nanoprecipitates in the Al matrix then cause a pinning effect on the dislocations, resulting in the accumulation of dislocations, which then leads to an increase in the dislocation density during the hot-rolling process.

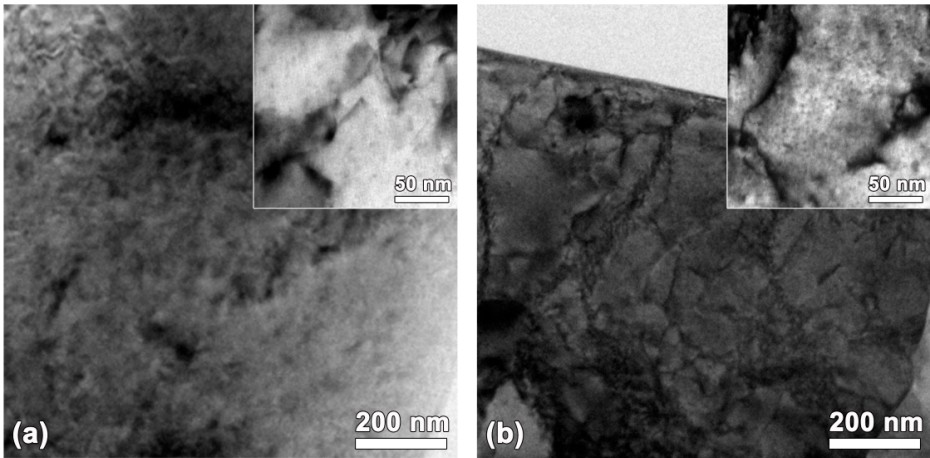

**Figure 6.** Typical BF images recorded from (**a**) sample A′ and (**b**) sample B′ with magnified BF images showing the effect of the amount of precipitation on the dislocation density after the hot-rolling process.

## 4. Conclusions

In this study, we demonstrated that the mechanical properties of Al5083 alloys can be effectively improved by the hot rolling and the trace addition of Sc. The highest tensile properties ($\sigma_{UTS}$ = 330 MPa, $\varepsilon$ = 12.4%) were achieved by the hot rolling and the trace addition of Sc (0.15 wt.%). It is, in general, accepted that mechanical strength and ductility are inversely proportional. The developed Al alloys, however, showed that the mechanical strength and the elongations were simultaneously improved. We then investigated the effect of Sc microalloying on the microstructural changes of the Al5083 alloy. A trace amount of added Sc affects the microstructure of the Al5083 alloy from the macroscopic to the microscopic scale in the following ways:

(1) Macroscopically, the Sc microalloying element significantly reduces the Al grain size from ~83.99 μm to ~41.29 μm;

(2) Microscopically, the Sc microalloying element forms $Al_3Sc$ nanoprecipitates. Moreover, the formation of other nanoprecipitates is significantly facilitated by the addition of the Sc element.

The increase in the number density of the nanoprecipitates then enhances the accumulation of dislocations during the hot-rolling process, leading to an enhancement of the work hardening during hot rolling. Therefore, the improvement of the mechanical properties of the Sc-added Al5083 alloy stems from grain refinement and the facilitated formation of nanoprecipitates.

**Author Contributions:** Alloy design and writing—original draft preparation, J.-H.P.; writing—review and editing, K.-A.L.; processing and methodology, S.-J.W.; data acquisition, Y.-B.K.; supervision, K.-H.K. All authors have read and agreed to the published version of the manuscript.

**Funding:** This research was financially supported by the Institute of Civil Military Technology Cooperation funded by the Defense Acquisition Program Administration and by the Ministry of Trade, Industry, and Energy of the Korean government under Grant No. UM18206RD2. This study was also partially supported by the R&D program from the Korea Institute of Industrial Technology.

**Institutional Review Board Statement:** Not applicable.

**Informed Consent Statement:** Not applicable.

**Data Availability Statement:** Data available on request.

**Conflicts of Interest:** The authors declare no conflict of interest.

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
