# Peer review of "Influence of Sc Microalloying on the Microstructure of Al5083 Alloy and Its Strengthening Effect"

_metals, doi:10.3390/met11071120_

Round 1

Reviewer 1 Report

Reviewer’s report on Park et al. „Influence of Sc microalloying …“

The paper by Park et al. discusses the interesting topic of Sc addition to aluminum alloys. The authors investigated specimen by the commercial alloy AA5083 in a given composition and one with 0.15 wt.% Sc addition. They applied tensile testing, XRD, optical microscopy and SEM together with EDX. 

The paper is well written mainly in good English. 

Generally, I had the problem that references 4 to 17 are missing in the text and 29 - 48 in the list of references. 

In detail, following points should be improved by the authors:

1. page 1 line 44-45

„The diffusivity of Mg atoms is then severely restricted by the Al3Sc precipitates.“

—> The authors should explain why they state this here, since no reference is given. 

2. page 2 line 47

„… finely dispersed Al3Sc precipitates reportedly inhibit recrystallization…“

—> the authors should give a reference here. This not obvious to the reader. 

3. page 4 line 123-125

„The Al3M type phase is generally a very fine size of few nanometers such that XRD is incapable of detecting Sc-related precipitates due to the relatively large beam probe size [25]“

—> a reference for this statement should be given here. 

4. page 5 line 176

„… those with the Al3M type structure are only a few nanometers in size, …“

—> the size of the precipitates is not relevant for XRD data, provided that is around 1nm or larger. The authors should explain what they mean by this statement. 

5. page 5 line 183-186

The reference to Fig. 5 A’ and B’ is obviously wrong. I guess it should read A and B. 

—> the authors should check this. 

6. Fig. 4 and 5

—> the legend is too small in the printed manuscript; this has to be improved - e.g. by splitting up these figure into two separate ones. 

After the given suggestions for minor revisions the paper should be published provided that the authors can present a correct PDF-File with all references given. 

Reviewer 2 Report

Thanks to the authors for the work done. however, there are a few notes on this article
1) please place links correctly in the introduction
2) the introduction also requires an extension and a more precise formulation of the problem. a lot of research has been done in this direction. how does your work differ from others in this direction?
3) it is not clear from the experimental procedure which samples were used for burst tests
4) How many samples did you test for each alloy? what is the confidence interval for each alloy tested?
5) the reasoning about the grain size is not convincing; in the presented figure 3 it is difficult to determine the grain size.
6) what is the size of the al3sc dispersoids? homogenization temperature 450 is negative for the growth and coherence of these particles. it would be interesting to look at these particles in high resolution TEM.
7) please describe the conclusions more specifically on the work done

Round 2

Reviewer 2 Report

1)The measurement error is usually of the same order of magnitude as the measurement. 330±10 or 330.1±10.3
2) please show the images where you measured the al3sc particle size. according to the images presented at the moment, the average size is difficult to determine, but the approximate size is 5-10 nm
